# Pressurized Liquid Extraction: A Powerful Tool to Implement Extraction and Purification of Food Contaminants

**DOI:** 10.3390/foods12102017

**Published:** 2023-05-16

**Authors:** Laura Barp, Ana Miklavčič Višnjevec, Sabrina Moret

**Affiliations:** 1Department of Agri-Food, Environmental and Animal Sciences, University of Udine, 33100 Udine, Italy; sabrina.moret@uniud.it; 2Faculty of Mathematics, Natural Sciences and Information Technologies, University of Primorska, Glagoljaška 8, 6000 Koper, Slovenia; ana-miklavcic@famnit.upr.si

**Keywords:** pressurized liquid extraction, food contaminants, sample preparation, accelerated solvent extraction, in-cell clean-up

## Abstract

Pressurized liquid extraction (PLE) is considered an advanced extraction technique developed in the mid-1990s with the aim of saving time and reducing solvent with respect to traditional extraction processes. It is commonly used with solid and semi-solid samples and employs solvent extraction at elevated temperatures and pressures, always below the respective critical points, to maintain the solvent in a liquid state throughout the extraction procedure. The use of these particular pressure and temperature conditions changes the physicochemical properties of the extraction solvent, allowing easier and deeper penetration into the matrix to be extracted. Furthermore, the possibility to combine the extraction and clean-up steps by including a layer of an adsorbent retaining interfering compounds directly in the PLE extraction cells makes this technique extremely versatile and selective. After providing a background on the PLE technique and parameters to be optimized, the present review focuses on recent applications (published in the past 10 years) in the field of food contaminants. In particular, applications related to the extraction of environmental and processing contaminants, pesticides, residues of veterinary drugs, mycotoxins, parabens, ethyl carbamate, and fatty acid esters of 3-monochloro-1,2-propanediol and 2-monochloro-1,3-propanediol from different food matrices were considered.

## 1. Introduction

Pressurized liquid extraction (PLE) was developed in the mid-1990s with the aim of saving time and reducing solvent with respect to traditional extraction processes. This technique is also called accelerated solvent extraction (ASE), pressurized fluid extraction (PFE), pressurized hot solvent extraction (PHSE), high-pressure solvent extraction (HPSE), high-pressure high-temperature solvent extraction (HPHTSE), and subcritical solvent extraction (SSE) [1].

Shortly after its first introduction, PLE was approved as an Environmental Protection Agency (EPA) method for the determination of semi-volatile organic compounds, organochlorine and organophosphorus pesticides, chlorinated herbicides, polychlorinated biphenyls (PCBs), polycyclic aromatic hydrocarbons (PAHs), and polychlorinated dibenzo-dioxins and -furans (PCDD/Fs) in solid and semi-solid environmental samples. Over the years, several applications of PLE have been developed, especially in the pharmaceutical, environmental, and food fields, for the extraction of bioactive and nutritional compounds and organic contaminants (i.e., veterinary drugs, pesticides, persistent environmental chemicals, and naturally occurring toxicants) [2].

The main aim of this review is to give an updated overview of PLE extraction techniques by describing the principles of the technique, instrumentation, and parameters affecting the extraction. In addition, the focus will be on the latest applications concerning sample preparation for food contaminant analysis.

## 2. Principle of the Technique and Advantages

PLE employs solvent extraction at elevated temperatures and pressures, always below the respective critical points, to maintain the solvent in a liquid state throughout the extraction procedure [1,3]. The physicochemical properties of the extraction solvent change under these particular pressure and temperature conditions, allowing easier and deeper penetration of the solid or semi-solid matrix to be extracted [1,4]. In particular, the solubility of analytes increases, the surface tension and viscosity of the solvent decrease, and at the same time, the mass transfer rate increases. These conditions result in a rapid extraction process with high yields and low solvent consumption [1]. In addition, the use of automated instruments allows the development of less labor-intensive methods and improves reproducibility.

The instrumental approach typically involves dispersing the sample with an inert material (e.g., drying agent or sand), placing the mixed sample in a steel extraction vessel, pumping the solvent, heating the vessel (usually 75–200 °C), and raising it to pressures around 100 atm. Solvent extraction can be performed statically, dynamically, or both, and the process may be repeated by increasing the number of extraction cycles, if necessary, to increase analyte recoveries. Finally, compressed gas is used to purge the sample extract from the cell into a collection vessel [1,3,5]. The resulting significant reduction in extraction time and solvent consumption has important economic and ecological implications due to lower disposal problems and limited diffusion of solvents into the atmosphere.

The PLE technique is equivalent to the most common solvent extraction techniques, such as traditional or automated Soxhlet, and, for many applications, provides results comparable to those achievable with microwave-assisted extraction (MAE). Although MAE also performs high-temperature and high-pressure extractions, PLE has the advantage of furnishing an already filtered extract that does not require additional steps to separate the solid residue from the sample. In addition, the ability to perform purification directly in the cell can make the extraction highly selective. In contrast, however, the procedure for preparing the cell is generally more labor-intensive and the instrumentation is more expensive [1,3,4]. Table 1 briefly summarizes the characteristics in terms of organic solvent consumption, process time, cost of instrumentation, advantages, and disadvantages of different traditional and modern techniques used for solid sample extraction.

## 3. Instrumentation

The instrumental requirements for performing a PLE process are relatively simple, so homemade or purpose-adapted extraction systems (e.g., coffee machines) can be used in addition to commercially available instruments. In any case, corrosion-resistant materials must be used, given the high pressures and temperatures typically employed [1].

Basically, the instrumentation consists of a solvent tank, a pump, an oven containing the extraction cells, several valves and restrictors, and collection vials. The basic instrumentation can vary depending on whether the process is static, in which a fixed volume of extractant is used, or dynamic, in which the extractant flows continually through the sample.

In either case, the solvent reservoir is connected to the high-pressure pump that introduces it into the extraction cells, where the sample to be analyzed, possibly mixed with dispersing agents or clean-up sorbents, is placed and helps to push out the extract once the process is finished. The extraction cells are placed in an oven to be heated, while the collection vials are located in the area below. Extraction pressure is controlled by special valves, and there may be an inert gas circuit (commonly nitrogen) that helps to remove solvent from the lines after static extraction. Dynamic PLE also requires a special high-pressure pump to more precisely control the solvent flow rate, solvent preheating coils, and a pressure limiter (back pressure regulator) instead of a static open/close valve used in the static system. A schematic PLE configuration for static and dynamic procedures is shown in Figure 1.

Two different types of PLE instruments are available on the market: one operating in series and the other in parallel. In the first case (Speed Extractor, Büchi, Flawil, Switzerland), the extraction cells, suspended vertically in a carousel, are automatically picked up one at a time by a self-sealing actuator with pneumatic movement and placed in the oven where they undergo solvent extraction. Between each sample, the line is rinsed by sending the solvent to the appropriate drain. The PLE system working in parallel (Dionex ASE, Thermo Fisher Scientific, Sunnyvale, CA, USA) is capable of processing multiple samples simultaneously, without mobile systems to move the cells in the oven.

### On-Line Coupling

Usually, commercial instruments work under static extraction conditions and are not suitable for on-line coupling with the analytical determination. However, a number of applications using laboratory-assembled equipment able to realize dynamic extraction have been described and have been used in coupled systems [6,7].

There are some PLE–solid phase extraction (SPE) applications for the simultaneous extraction and purification of compounds from different foods, achieving high selectivity and separating them into classes without compromising extraction yield, even when using green solvents [8]. Furthermore, PLE can be coupled to analytical tools such as chromatography, allowing the on-line detection of target compounds [9].

Several applications involving on-line coupling of pressurized hot water extraction (PHWE) with either liquid chromatography (LC) or gas chromatography (GC) have been reported [6]. In this case, the extraction solvent is water at temperatures higher than 150 °C. PHWE is an ecofriendly method widely used to extract bioactive compounds [10,11]. By increasing the temperature of water, its dielectric constant decreases, which weakens the hydrogen bonds and makes water similar to less polar organic solvents such as methanol and ethanol. By reaching temperature of over 200 °C, extraction of apolar contaminants is also possible [12]. PHWE can also be performed in dynamic mode, in which water is continuously passed through the extraction cell. After extraction, the water is cooled, and the extracted analytes are collected in a solid-phase trap (for LC or GC) or in a membrane extraction unit (for GC). The extract from the sold-phase trap can be eluted directly on the LC column with a suitable eluent, while in PHWE-GC the trap must be dried with a gas stream before eluting the analytes [6].

## 4. Optimization of the Extraction Process

The efficiency of the extraction depends on the nature of the sample matrix, the analyte to be extracted, the interactions of the analyte with the matrix, and its distribution. To achieve optimal extraction yields and high selectivity in a PLE procedure, there are several parameters to be optimized.

### 4.1. Sample Pretreatment, Dispersing Agents, and Extraction Solvent

Homogenization is an important step in sample preparation to ensure that a representative sample is taken. Pretreatment usually involves sieving or grinding the sample because the diffusion of analytes from the sample to the solvent extract can be increased considerably by decreasing the particle size. In fact, the greater the surface area exposed to the solvent, the faster the extraction process will be. However, some matrices, when reduced to very fine particles, tend to give rise to packed masses that tend to obstruct the passage of solvent as extraction proceeds. For this reason, almost all applications of PLE involve the use of a dispersing agent (usually quartz sand or diatomaceous earth, DE), which is intended to prevent aggregation of sample particles by increasing the surface area of the sample exposed to solvent contact, to make packing homogeneous so as to avoid the formation of preferential pathways by the solvent and to fill empty volumes by reducing solvent consumption [4].

Samples with a high water content, which can prevent non-polar solvents from reaching the analytes within the matrix, are usually subjected to oven drying or freeze-drying before extraction, or previously purified drying agents (sodium sulfate, Na_2_SO_4_) are added. Extraction of wet samples using water as the extraction solvent can be an effective approach because, in some cases, it eliminates this drying step, minimizing sample pretreatment [4].

To perform an effective extraction, the solvent must be able to facilitate the release of the analyte of interest from the matrix and solubilize it, possibly without extracting other interfering compounds from the starting sample. The polarity of the solvent should be close to that of the target compound. Mixing the solvent with some organic and inorganic solvents, surfactants, and additives can facilitate the solubility of analytes in the extraction step, as well as affect the physical properties of the matrix and the desorption of analytes in the extraction step [2,13,14]. The choice of extraction solvent should also consider compatibility with subsequent treatments such as clean-up of the extract or analytical technique, as well as the volatility of the solvent if the concentration of the extract is required [4].

### 4.2. Temperature

Temperature is generally the most important factor to optimize an extraction process, as it is able to influence speed, efficiency, and selectivity [3]. The temperature increase promotes the extraction process by reducing the intermolecular interactions between the analyte and the matrix, acting on van der Waals forces, hydrogen bonds, and dipole–dipole interactions [9,12]. In addition, an increase in temperature changes solvent properties such as viscosity, diffusivity, and surface tension, resulting in a fast mass transfer (diffusion rate) and improved wetting of the sample [8,13]. However, an increase in temperature not only increases the solubility of the analyte but can also increase the solubility of other not desired compounds in the matrix, thus penalizing the selectivity of the extraction [9]. Moreover, the use of high temperatures must be carefully evaluated when the target compounds are thermolabile, as degradation of the analyte may occur during the extraction process [1,13]. In this case, thermal degradation can be decreased by using lower pre-heating and extraction times.

### 4.3. Pressure

Elevated pressures of 500–3000 psi (35–200 bar) are used to keep the solvent in liquid form even above its boiling temperature [13]. Pressure also has the effect of promoting penetration of the solvent into the pores of the matrix, forcing the solvent into areas that would not normally be contacted using atmospheric conditions, facilitating the extraction of the analyte, and promoting solubilization of air bubbles present within the matrix that hinder analyte–solvent contact [2].

### 4.4. Extraction Time and Number of Cycles

Extraction time is defined as the time the solvent is in contact with the matrix at the desired pressure, temperature, and flow rate and is optimized based on the matrix, analyte of interest, and extraction mode (static or dynamic) [1]. In the static mode, long exposure to solvents allows the matrix to swell and improve solvent penetration into the sample. In this case, extraction efficiency strongly depends on the equilibrium partitioning constant and solubility of compounds. If low recoveries are obtained in a single step, the static process can be repeated several times by sending a new aliquot of fresh solvent into the cell [2].

In the dynamic mode, a continuous flow of the extraction solvent at an appropriate rate through the cell is performed, allowing a short contact time between the sample and the solvent, thus improving mass transfer. However, this type of extraction is rarely used, mainly because of solvent consumption compared with the static process [2,12].

### 4.5. Effect of Other Parameters

Preheating time, flush volume, and purge time are three other parameters that need to be optimized even though they do not significantly affect sample recoveries. The preheating time is the time the extraction cell containing the sample is kept in the oven at the selected temperature before the solvent is added. Usually, 5 min is sufficient to ensure that the cell reaches the preset temperature [5]. Flash volume is the percentage of fresh volume introduced into the cell after the static time to transport the analytes to the collection vial. This additional solvent volume ensures the elution of all analytes, but it also increases the final volume [2].

### 4.6. Selective PLE

As previously reported, extraction selectivity can be improved by proper optimization of the extraction solvent and extraction temperature. Since PLE often leads to co-extraction of interfering compounds, an additional purification step on a SPE cartridge is usually required prior to the analytical determination [4]. Many methods, called in-cell clean-up, on-line clean-up, or selective PLE (S-PLE), combine the extraction and clean-up steps by including a layer of an adsorbent that retains interfering compounds (usually fats) directly in the PLE extraction cells [4,12,13]. In this case, when the pressurized solvent is eluted, part of the matrix components and other interferences are retained in these solid phases, and, at the same time, elution of the analytes is carried out. The most widely used materials for the in-cell clean-up are Florisil, silica in different versions (acidic, basic, neutral, activated or not, treated with silver nitrate or copper), and alumina [13].

Examples of the use of this approach are given in the following sections.

## 5. Applications in Food Contaminants

Table 2, Table 3 and Table 4 show the most relevant applications of PLE published in the past 10 years related to the extraction of different contaminants from food. The range of compounds analyzed includes persistent organic pollutants (POPs), endocrine-disrupting compounds (EDCs), flame retardants (FRs), hydrocarbon contaminants (PAHs, heterocyclic aromatic amines—HCAs, and mineral oils—MOHs) (Table 2). In addition, PLE has been widely used both for selective and multi-residue extraction of different classes of pesticides and veterinary drugs along with related metabolites (Table 3). Applications in other contaminants such as aflatoxins, mycotoxins, ethyl carbamate, and fatty acid esters of monochloropropane (MCPDEs) are also reported (Table 4). PLE has proven to be a technique that can be easily combined with different types of determination systems (LC, GC, capillary electrophoresis—CE).

### 5.1. Environmental and Processing Contaminants

#### 5.1.1. EDCs

EDCs are capable of disrupting or changing the usual functions of the endocrine system. They can be accumulated in fatty matrices; therefore, their contents in food samples should be accurately determined. Common EDCs are bisphenol A (BPA), dioxins, perchlorate, perfluoroalkyl and polyfluoroalkyl substances, phthalates, phytoestrogens, polybrominated diphenyl ethers (PBDEs), PCBs, and triclosan.

Given their apolar nature, many lipophilic contaminants are found in particularly high concentrations in high-fat foods. PLE has been successfully applied in these matrices using apolar solvents (hexane or cyclohexane) and combinations with medium polarity solvents such as dichloromethane or acetonitrile. Additional clean-up is required to remove co-extracted lipids. Off-line extract clean-up involves SPE using conventional or custom cartridges with different retention phases and gel permeation chromatography (GPC).

**Table 2 foods-12-02017-t002:** Application of PLE for the analysis of environmental and processing contaminants in food.

Analytes	Matrix	Sample Pre-Treatment	Cell Preparation	Extraction Solvent	In Cell Clean-Up	T(°C)/Pressure	Time/No. Cycles/Flush Volume	Purge with Nitrogen	Sample Post-Treatment	Analysis	Ref.
PCDD/Fs	Fish	Ground and freeze-dried	Sample (equivalent to 20 g of wet weight) in a 34 mL cell	Hexane	no	100/18 kPa	15 min/3 cycles/90%	120 s	SPE (silica gel)	GC-HRMS	[14]
PCBs	Fish	Freeze-dried	1 g of sample in a 33 mL cell mixed with DE	Acetone/hexane (1/1, *v*/*v*)	no	100/10 MPa	5 min/1 cycle/60%	90 s	SPE (Extrelut-NT3)	GC-MS	[15]
PCDD/Fs, dl-PCBs	Clam and crab tissues	Triturated	10 g of sample mixed with anhydrous Na_2_SO_4_ in a 100 mL cell and mixed with 5 g alumina, 5 g celite, 0.8 g carbopack, 5 g Florisil, and 5 g of silica gel	Dichloromethane/hexane (1/1, *v*/*v*) for PCBs or toluene for PCDD/Fs	yes	100/1500 psi	5 min/1 cycle for PCBs and 1 subsequent cycle for PCDD-Fs/75%		Concentrated	HRGC–ECNI/MS	[16]
PBDEs, PCBs, OCPs	Bowhead whale blubber	Homogenized	1.5 g of sample (wet weight) mixed with anhydrous Na_2_SO_4_ in a 100 mL cell with 55 g of acidic silica and 5 g of baked neutral silica	Hexane	yes	100/1500 psi	5 min/2 cycles/100%		Concentration	GC-MS	[17]
PBDEs	Fish	Freeze-dried and pulverized	1 g of sample in a 22 mL cell mixed with 3 g acid-washed sand, 1 g DE, and 2 g activated silica gel	Hexane/dichloromethane (1/1, *v*/*v*)	yes	100/1500 psi	5 min/3 cycles/60%	120 s		GC–QqQMS	[18]
POPs	Fish	Freeze-dried and ground	2 g of sample mixed with 10 g Na_2_SO_4_ in a 33 mL cell	Dichloromethane/hexane (1/1, *v*/*v*)	no	100/10.3 MPa	5 min/3 cycles		SPE (multilayer silica gel, alumina, and Florisil)	HRGC-HRMS or HRGC-NCI-LRMS	[19]
PCBs	Chicken, clam, pork meat	Homogenized	1 g of sample in a 33 mL cell mixed with alumina, Florisil, silica gel, Celatom, and copper(II) isonicotinate	Hexane	yes	120/1500 psi	15 min/1 cycle	120 s	Concentration	GC-MS	[20]
OH-PBDEs	Crayfish and grass carp	Blended	5 g of sample mixed with Florisil (1/2, *w*/*w*)	Cyclohexane/ethyl acetate (1/1, *v*/*v*)	no	100/1500 psi	5 min/3 cycles	90 s	GPC (S-X3 Bio-Beads)	LC-MS/MS	[21]
diOH-PBDEs	Sea fish	Freeze-dried and ground	5 g of sample mixed with 2 g of diatomite with 1 g quartz sand, 5 g neutral silica gel, and quartz sand	Dichloromethane/methanol (1/3, *v*/*v*)	yes	100/100 bar	15 min/3 cycles		Partitioning with a KOH solution, SPE (Florisil), and derivatization	GC-MS/MS	[22]
OPs	Wild boar liver		2 g of sample mixed with 7.5 mL KOH (60%, *w*/*v*), 35 g activated silica, and 1.0 g of anhydrous Na_2_SO_4_	Acetonitrile	yes	100/150 bar	10 min/3 cycles		SPE (dual-layer EZ-POP)	GC-QqQ-MS/MS	[23]
PAEs	Seafood species	Freeze-dried, homogenized, and sieved to 500 μm	1 g (dry weight) of sample in a 11 mL cell mixed with 1.6 g of DE	Methanol	no	80/1500 psi	10 min/1 cycle/60%	90 s	SPE (Bond Elute Plexa)	LC-HRMS	[24]
EDCs (BPA and APs)	Clams	Freeze-dried	0.5 g of sample mixed with 3 g of neutral alumina and silica gel	Methanol	yes	40/1500 psi	10 min/1 cycle/60%		Concentration and dissolution in 1 mL of methanol	LC-MS/MS	[25]
APs, BPA	Wild mussels	Freeze-dried	0.5 g of samples mixed with 1.5 g of silica in a 11 mL cell with 3 g neutral alumina (5% water deactivated)	Methanol	yes	40/10 MPa	5 min/2 cycles/60%		Evaporation to dryness and dissolution in 1 mL of methanol	LC-MS/MS	[26]
EDCs	Cheese		5 g of sample in a 22 mL cell with 1.5 mL acetone and 50 μL carbon tetrachloride	Acetone	no	70/1000 psi	5 min/1 cycle		DLLME	HPLC-DAD	[27]
MOHs	Dry foods: semolina pasta, rice, and cereals	Ground for total contamination	8 g of whole sample in a 10 mL cell or 2 g of ground sample in a 10 mL cell mixed with 5 g of fat free quartz sand and 6 g of sand	Hexane or hexane/ethanol (1/1, *v*/*v*)	no	100/100 bar	5 min/1 cycle + 2.5 min at 2 mL/min or 5 min/2 cycles	60 s	−20 °C for about 20 min	LC-GC-FID	[28]
MOHs	Dry semolina pasta		8 g of sample in a 10 mL cell	Hexane	no	100/100 bar	5 min/1 cycle + 2.5 min at 2 mL/min	60 s	−20 °C for about 20 min	LC-GC-FID	[29,30]
PAHs	Roasted coffee	Homogenized by quartering	10 g of sample in a 66 mL cell with 16 g of activated silica gel	Hexane/dichloromethane (85/15, *v*/*v*)	yes	100/10.34 MPa	5 min/2 cycles/100%	50 s	LLE and SPE (silica gel)	GC-MS	[31]
PAHs	Cereal based foods	Ground to a fine powder	5 g of sample in a 33 mL cell mixed with 5 g of polyacrylic acid and 15 g of pre-cleaned Ottawa sand	Hexane	no	100/10 MPa	10 min/2 cycles/60%	120 s	SPE (silica gel)	GC-MS	[32]
PAHs	Fish	Freeze-dried and ground	0.01–0.03 g of sample in a 11 mL cell with 5 g silica gel and sand	Hexane/dichloromethane (9/1, *v*/*v*)	yes	100/1500 psi	5 min/1 cycle/60%	60 s	Concentrated	GC-MS	[33]
PAHs and oxyPAHs	Mussels	Dried with Na_2_SO_4_	25 g of sample in a 22 mL cell mixed with DE	Hexane/acetone (3/1, *v*/*v*)	no	100/10.3–11.7 MPa	5 min/3 cycles/60%	60 s	SPE (silica gel)	LC-MS	[34]
BaP metabolites	Liver tissue	Cut in small pieces and homogenized	1 g of sample in a 10 mL cell with 10 g of activated Florisil, DE, and Ottawa sand	Methanol/chloroform/water (30/15/10, *v*/*v*/*v*)	yes	100/10 psi	10 min/2 cycles		Evaporation and dissolution in 0.5 mL of methanol	HPLC-FLD and UHPLC-APCI-MS/MS	[35]
PAHs	Smoked bacon	Ground and homogenized	5 g of sample mixed with 2 g of DE in a hard-cap coffee machine	Water/acetonitrile (80/20, *v*/*v*, with 0.1% formic acid)	no	75/19 bar	10–15 s		Addition of Florisil (200 mg)	HPLC-MS/MS	[36]
PAHs	Seafood	Homogenized using a mixer	2 g of sample mixed with 2 g of DE in a hard-cap coffee machine	Water/acetonitrile (60/40, *v*/*v*)	no	75/19 bar	10–15 s		Addition of NaCl and MgSO_4_, evaporation to dryness, and dissolution in 200 µL of acetonitrile	LC-APCI-MS/MS	[37]
HCAs	Cooked meat products	Homogenized in a high-speed food blender	5 g of sample dissolved in 12 mL of 0.5 M NaOH (70/30 methanolic/aqueous solution) and mixed with 12 g of DE in a 66 mL cell with 10 g of neutral alumina	Dichloromethane/acetonitrile (1/1, *v*/*v*)	no	80/10.3 MPa	5 min/2 cycles/50%	160 s	Evaporation to dryness and dissolution in 1 mL of 30 mM formic acid– acetonitrile (90/10, *v*/*v*)	LC-MS-IT-TOF	[38]
PFRs	Fish	Freeze-dried and triturated	1 g of sample in a 33 mL cell mixed with 2 g of acid-washed silica gel and DE	Acetonitrile/water (9/1, *v*/*v*)	yes	150/1500 psi	5 min/1 cycle/60%	300 s	SPME	GC-FPD	[39]
BFRs	Fish, crustaceans, milk, eggs, muscle, and sheep liver		Sample corresponding to 1 g of fatty extract in a 34 mL cell	Toluene/acetone (70/30, *v*/*v*)	no	120/100 bar	5 min/3 cycles/81–100%	150 s	SPE (acidified silica, Florisil, and carbon)	GC-EI-HMRS and LC-HMRS	[40]
FRs	Elephant seal (*Mirounga leonina*) and Antarctic fur seal (*Arctocephalus gazella*)	Freeze-dried and ground with alumina (1:2)	1.5 g of sample in a 22 mL cell with 6 g of alumina and DE	Hexane/dichloromethane (1/1, *v*/*v*)	no	100/1500 psi	10 min/2 cycles/80–100%		SPE (neutral alumina)	GC-MS/MS	[41]

S-PLE with adsorbents capable of retaining fats has been largely used for the extraction of PCBs and PBDEs from different food matrices. The analysis of PCDDs/Fs and dioxin-like polychlorobiphenyls (dl-PCBs) congeners is complicated by the low concentration at which these contaminants are present in food compared to non-dl-PCBs (present at much higher levels). The traditional procedure for the determination of these contaminants from complex matrices such as food involves the extraction of fat with Soxhlet, followed by multistep purification on different multilayer columns to eliminate potential interferents, and fractionation/isolation of planar compounds (of greater toxicological relevance) from non-planar PCBs (not similar dioxin). Because of all these steps, the analysis is particularly long and expensive. The development of S-PLE methods allowed the analysis of these contaminants in food matrices involving fat removal directly in the extraction cell (in the presence of clean-up sorbent) in order to obtain extracts suitable for direct GC analysis [16,17,18,20] or further purification steps (SPE) [22,23].

Phthalates (phthalic acid esters, PAEs) are a group of high-production chemicals used in plastic manufacturing primarily to increase the flexibility of plastics such as polyvinyl chloride. These compounds are found in food packaging and food supplements, personal care products, textiles, and medical equipment. Because these compounds are not chemically bound to plastic products, they can easily contaminate the environment by reaching the air and water and even entering food products. In the past years, numerous methods have been developed for determining phthalates and, to a much lesser extent, their metabolites in different matrices, including extraction procedures such as QuEChERS and ultrasound extraction. Hidalgo-Serrano and coworkers [24] proposed a PLE-based method in which freeze-dried seafood samples were mixed with DE as dispersing agent and then extracted with methanol at 80 °C (1 cycle, 10 min) prior to the SPE and LC–high-resolution mass spectrometry (HRMS) determination. Because of the ubiquitous distribution of these contaminants, laboratory analysis is particularly complicated. In fact, glassware must be properly rinsed with phthalate-free solvents and allowed to dry completely before use. Environmental contamination must also be avoided during sample preparation. Concerning PLE, procedural blanks performed with DE instead of fish resulted in the identification of mono(2-ethylhexyl) phthalate and dibutyl phthalate (<10 μg/L), and diethyl phthalate and bis(2-ethylhexyl) phthalate (<50 μg/L). The corresponding blank values were then subtracted from the analyzed samples to ensure that environmental contamination did not lead to false positives.

Alkylphenols (APs), degradation products of the non-ionic surfactants alkylphenol polyethoxylates, are used as plasticizers in high-density polyethylene, polyethylene terephthalate, and polyvinyl chloride and also in the manufacture of textiles, paper, and agricultural chemical products. Another known important EDC is BPA, used as a monomer for the production of epoxy resins, phenol resins, polycarbonates, polyesters, and lacquer coatings for food cans. The analysis of these two EDCs with a PLE-based method has been proposed by Salgueiro-González and coworkers [25,26] and applied to biota and mussel samples, in order to monitor their occurrence, distribution, and bioaccumulation. In this method, freeze-dried samples were extracted by S-PLE with neutral alumina as a clean-up sorbent and methanol as extraction solvent at 40 °C prior to the LC–tandem mass spectrometry (MS/MS) determination.

Multiresidue analysis of EDCs (including di-butyl phthalate, di-iso-nonyl phthalates, di-ethylhexyl phthalate, di-ethylhexyl adipate, and BPA) in cheese samples was recently published by Pil-Bala and coworkers [27]. The authors first developed a PLE dispersive liquid–liquid microextraction (DLLME) method in which the solid samples were firstly extracted in PLE procedure using a water-miscible extraction solvent (acetone, at 70 °C in the presence of carbon tetrachloride as a modifier), and then the solvent was used as a dispersive solvent in the following DLLME procedure.

#### 5.1.2. Hydrocarbon Contaminants

MOHs are complex mixtures of saturated and aromatic hydrocarbons of petrogenic origin that can contaminate food through various sources. An important source of contamination is represented by recycled cardboard packaging (produced from wastepaper containing residues of printing ink), which transfers the most volatile part of its contamination to the food it packs. In 2014, a PLE method was developed for the rapid extraction of these contaminants from dry foods such as pasta and cereals. In particular, by exploiting the different selectivity of the solvent, two methods have been developed: one for the extraction of surface contamination migrated by packaging and the other for the extraction of total contamination from different sources [28]. In the first case, the extraction is carried out with hexane at 100 °C (5 min, one cycle) on the whole product or coarsely chopped (without the use of dispersing agents), while in the second case, the extraction (2 cycles of 5 min at 100 °C) is carried out on the ground sample (dispersed with quartz sand), using a mixture of hexane/ethanol (1/1, *v*/*v*), which has the ability to swell the starch and denature the proteins resulting in a quantitative release of the contaminants already present in raw materials, which otherwise are not accessible to the apolar solvent. The first method was then applied by Barp and coworkers [29,30] to evaluate hydrocarbon contaminants in dry semolina and egg pasta migrating from different packaging materials (virgin and recycled paperboard and polypropylene film).

PAHs are an important class of organic contaminants that contain two or more fused aromatic rings and result from the incomplete combustion of organic matter. Their presence in food, deriving from environmental contamination, technological processes, or contaminated packaging materials, poses a potential risk to human health. Fat extraction and clean-up steps are required for the determination of PAHs in complex matrixes. S-PLE has been proposed in the literature as an innovative technique to perform these steps simultaneously, with automation and reduced time and solvent consumption [31,33,35]. When activated silica gel is used as an adsorbent material, a significant number of heavier PAHs can be retained within it without being extracted, but careful optimization of the hexane volume in the extraction solution can increase recoveries. A hexane/dichloromethane mixture (85/15, *v*/*v*) at 100 °C (2 cycles, 5 min) was applied for the extraction of PAHs from roasted coffee [31], while hexane/dichloromethane (9/1, *v*/*v*) at 100 °C (1 cycle, 5 min) was used in fish samples [34]. Benzo[a]pyrene (BaP) metabolites were analyzed in liver tissue using S-PLE with Florisil, a mixture of methanol/chloroform/water (30/15/10, *v*/*v*/*v*) at 100 °C (2 cycles, 10 min) [35].

Other authors have preferred to use an off-line clean-up, subjecting the PLE extract to SPE on silica gel prior to analytical determination (GC- or LC-MS) of PAHs and their oxygenated derivatives (oxyPAHs) in cereal-based foods and mussels, respectively [32,34].

An alternative PLE using a hard-cap espresso machine was applied for the determination of PAHs in smoked bacon [36] and in seafood [37] by LC-MS/MS. Appropriately pretreated samples (ground and homogenized) were mixed with dispersing agents (DE) and transferred to an extraction capsule. A hard-cap coffee machine with an operating pressure of 19 bar was used for the extraction using a water/acetonitrile mixture in the extraction solution at 75 °C (10–15 s). The extracts were then cleaned-up by the addition of Florisil or QuEChERs. In these applications, the methods developed proved to be efficient, reliable, fast, and cheap.

HCAs are a large class of different substances with high mutagenic and carcinogenic potential resulting from the heating of protein-rich foods. Usually, analytical procedures may involve a series of purification and pre-concentration steps using large amounts of organic solvents, followed by various separation and detection techniques. With the aim of obtaining higher extraction efficiencies consuming less solvent and labor time, PLE was proposed for HCAs in food samples by Ouyang and coworkers [38]. In particular, raw meat products (chicken breast, duck breast, pork fillet, and bream loin) roasted (230 °C for 20 min) or fried (200 °C for 20 min) were then extracted with dichloromethane/acetonitrile (1/1, *v*/*v*) at 80 °C (2 cycles, 5 min). Neutral aluminum oxide was used to remove grease, pigments, and other impurities, then the extract was analyzed directly by LC-MS/MS.

#### 5.1.3. FRs

FRs are compounds that are applied to materials to increase their fire resistance. PBDEs are the most used family in a wide variety of indoor and outdoor products, such as household appliances, office electronics, textiles, and furniture. Due to the restriction on the use of PBDEs, organophosphorus flame retardants (PFRs) have been extensively used for several decades, and their consumption may greatly increase in the future. Their determination in seafood by PLE has been reported in the literature. Gao and coworkers [39] proposed a S-PLE approach using aqueous solutions (water/acetonitrile, 90/10, *v*/*v*) at 150 °C for 5 min and acid-washed silica gel used as lipid sorbent followed by solid phase microextraction (SPME) and use of a GC-flame photometric detector (FPD). FRs have been analyzed in seafood also by PLE and subsequent SPE on different materials (acidified silica, Florisil, carbon, and neutral alumina) using organic solvents to prove the long-range transport capacity of these contaminants and their widespread diffusion in food samples [40,41].

### 5.2. Pesticides and Residue of Veterinary Drugs and Anabolic Substances

#### 5.2.1. Pesticides

Different fungicidal compounds (metalaxyl, cyprodinil, procymidone, iprovalicarb, myclobutanyl, kresoxim-methyl, benalaxyl, fenhexamide, tebuconazole, iprodione, and dimethomorph) were efficiently extracted from white grape bagasse by using PLE followed by GC–triple-quadrupole mass spectrometry (QqQ-MS). The optimized method involves 80 °C for 15 min, and extraction was performed using hexane/acetone (1/1, *v*/*v*). A direct comparison with an ultrasound-assisted extraction method developed in parallel revealed better performance with the PLE approach, which provided significantly superior responses [42].

Screening of priority pesticides, a group of banned and toxic substances known to persist in the aquatic environment, was conducted in *Ulva* sp. algae by GC coupled with electron capture detection (ECD). Extraction and clean-up of the samples were performed in one step by S-PLE with Florisil. For most of the 21 compounds studied, peak areas increased with the increase in temperature from 80 to 120 °C and decreased with the increase in pre-heating time. Better performance was obtained with S-PLE than with the traditional and more widely used Soxhlet extraction, not only in terms of recoveries but also considering the reduction in analysis time [43].

S-PLE was used also for the extraction and clean-up of organochlorine pesticides (OCPs) in fish. The lipid-removal efficiencies achieved by adding alumina, Florisil, acid-treated silica gel, and silica gel to the extraction cell were compared. A higher lipid content was observed when two adsorbents were used together than when only one adsorbent was used. Thus, in the optimized method, fish (2–3 g) was placed above alumina (30 g) in the extraction cell, then the sample was extracted using a mixture of hexane/dichloromethane (7/3, *v*/*v*) at 100 °C. Advantages of the S-PLE include a short preparation time, minimal sample contamination, the use of little solvent, and the ability to be automated. For these reasons, the S-PLE has proven suitable for use in environmental and food industries, which require rapid analyses of contaminants [44].

**Table 3 foods-12-02017-t003:** Application of PLE for the analysis of pesticides and veterinary drugs in food.

Analytes	Matrix	Sample Pre-Treatment	Cell Preparation	Extraction Solvent	In Cell Clean-Up	T (°C)/P	Time/No. Cycles/Flush Volume	Purge with Nitrogen	Sample Post-Treatment	Analysis	Ref.
Fungicides	White grape bagasse	Dried and pulverized	0.5 g of sample in a 10 mL cells with 1 g of clean sand	Hexane/acetone (1/1, *v*/*v*)	no	120/1500 psi	5 min/1 cycle/60%	60 s		GC-MS	[42]
Pesticides	Seaweeds	Triturated	5 g of sample mixed with 10 g of anhydrous Na_2_SO_4_ and 1 g of DE in a 33 mL cell with 2.5 g of Florisil and 0.5 mL of ethylacetate	Hexane	yes	120/1500 psi	5 min/1 cycle/60%	150 s	Evaporation to 0.5 mL	GC-ECD	[43]
OCPs	Fish tissue	Homogenized and freeze-dried	2–3 g of sample mixed with DE (1:2) in a 66 mL cell with 1.5 g DE and 30 g of alumina	Hexane/dichloromethane (7/3, *v*/*v*)	yes	100/1500 psi	5 min/2–3 cycles/60%	100 s		GC-MS	[44]
Pesticides	Tuber crops		3 g of sample (0.5% acetic acid) in a 10 mL cell mixed with DE	Ethyl acetate	no	100/1400 psi	5 min/3 cycles/60%	60 s	Evaporation and dissolution in 1 mL ethyl acetate	GC-MS/MS	[45]
Fungicides	Matcha		1 g of sample in a 66 mL cell with 2 g of C18, 4 g Florisil and 30 g Anasorb 747	Ethyl acetate	yes	100/1500 psi	30 min/4 cycles/60%	600 s	Evaporation	LC-ESI^+^-MS/MS	[46]
Pesticides	Leaves	Homogenized with a mortar and pestle	0.5 g of sample mixed with DE in a 34 mL cell with 5 g of Florisil, 0.6 g of GCB and sand	Ethyl acetate/hexane (25/75, *v*/*v*)	yes	80/1500 psi	10 min/3 cycles/50%	120 s	Evaporated	GC-MS	[47]
Pharmaceuticals	Fish		1 g of sample in a 22 mL cell mixed with DE and 2 g of neutral aluminum oxide	Methanol	yes	50/1500 psi	5 min/4 cycles		Evaporation to dryness and dissolving in organic solvent. Further sample purification (SPE Florisil, GPC, and SPE followed by GPC)	UHPLC-MS	[48]
TCs	Chicken eggs, muscle of fish, and shrimp		5 g of sample in a 22 mL cell mixed with 5 g of Na_2_EDTA-washed sand	TCA/methanol (1/3, *v*/*v*)	no	60/65 bar	3 min/2 cycles/80%	60 s	Evaporation to dryness, dissolution in 1 mL of mobile phase	HPLC-UV	[49]
Pharmaceuticals	Cooked and uncooked marine blue mussels	Freeze-dried, homogenized, and sieved to 125 μm	1 g of sample mixed with 10 g of Ottawa sand in a 33 mL cell with 20 g of activated neutral aluminum oxide and sand	Acetonitrile/water (3/1, *v*/*v*)	no	60	5 min/3 cycles		SPE (Strata-X)	LC-MS/MS	[50]
Pharmaceuticals	Blue mussels	Freeze-dried and sieved to 125 μm	1 g of sample mixed with 10 g of Ottawa sand in a 33 mL cell with 20 g of activated neutral aluminum oxide and sand	Acetonitrile/water (3/1, *v*/*v*)	no	60	5 min/3 cycles		SPE (Strata-X)	LC-MS/MS	[51]
Pharmaceuticals and metabolites	Bivalves	Freeze-dried and ground	0.5 g of sample (dry weight) mixed with DE in a 22 mL cell with 2 g of neutral aluminum oxide	Methanol/water (1/2, *v*/*v*)	no	50/1500 psi	5 min/3 cycles		SPE (Oasis HLB)	HPLC–QqLIT	[52]
Sulfonamides	Samples of ovine (muscle, liver, and kidney), poultry (liver), equine (liver), and fish (muscle)	Freeze-dried and ground	0.5 g of sample mixed with DE	Hexane (clean-up); acetonitrile with 0.2% acetic acid (extraction)	yes	60/1500 psi (clean-up); 90/1500 psi (extraction)	5 min/2 cycles/80% (clean-up); 7 min/3 cycles/80% (extraction)	60 s (extraction)	−18 °C for 1 h, evaporation and dissolution in 1 mL of mobile phase	HPLC–QqLIT-MS/MS	[53]
TCs	Chicken meat, clam meat, and pork samples	Homogenized in a high-speed food blender	3 g of sample mixed with 3 g copper(II) isonicotinate powder in a 33 mL cell	Methanol	yes	70/1500 psi	15 min/1 cycle	120 s	Evaporation	LC–MS/MS	[54]
Amphenicols residues	Poultry tissues	Ground	2.5 g of sample mixed with 2 g DE in a 11 mL cell	0.2% ammonium hydroxide in water	no	150/100 bar	3 min/2 cycles		SPE (Oasis HLB)	UPLC-MS/MS	[55]
Pharmaceuticals	Mussels	Freeze-dried, homogenized, and sieved to 125 μm	1 g of sample in a 11 mL mixed with 3 g of Ottawa sand	Ultrapure water	no	100/1500 psi	10 min/1 cycle/150%	300 s	SPE (Oasis MAX)	LC-MS/MS	[56]
Pharmaceuticals and metabolites	Wild fish	Freeze-dried and ground	0.5 g of sample	Methanol	no	50 °C	5 min/4 cycles		GPC	HPLC-MS/MS	[57]
Antibiotics	Shrimp and sardine	Blended and dried	5 g of sample mixed with 1.5 g of diatomite in a 33 mL	Acetonitrile	no	60/10.3 Mpa	5 min/2 cycles/40%	90 s	Evaporation to dryness and dissolution in 1 mL of methanol	CE	[58]
Antibacterial agents	Fish	Ground	1 g of sample mixed with 3 g of sea sand in a 11 mL cell with sea sand	Water	no	100/1500 psi	15 min/1 cycle/60%	60 s	SPE (Absolute Nexus)	LC-ESI-MS	[59]
Pharmaceuticals	Fish	Freeze-dried and ground	1 g of sample in a 22 mL cell mixed with DE and 2 g of neutral aluminum oxide	Methanol	no	50/1500 psi	5 min/3 cycles		GPC	UHPLC-MS	[60]
Veterinary drugs	Fatty foods (shrimp, crab meat stick, salmon, lobster, chicken liver, pork sausage)	Freeze-dried and ground	3 g of sample mixed with 3 g DE in a 34 mL cell with sea sand	Hexane/acetonitrile/methanol (2/1/1, *v*/*v*/*v*)	yes	60/1500 psi	10 min/2 cycles/105%	300 s	Concentration and dissolution in 10 mL of solvent	UHPLC-MS/MS	[61]
Veterinary drugs	Chicken	Freeze-dried, homogenized, and sieved	3 g of sample mixed with 3 g of DE in a 34 mL cell with sea sand	Water	no	60/1500 psi	5 min/1 cycle/100%	100 s	Precipitation of proteins by adding acetonitrile	UHPLC-MS/MS	[62]
Pesticides	Aquatic animals	Freeze-dried and homogenized	0.5 g of sample mixed with DE (5/1) and 0.7 g of Florisil in a 5 mL cell	Methanol	yes	40/1500 psi	3 min/3 cycles/60%	60 s	SPE (on-line)	UPLC-ESI-MS/MS	[63]

For the extraction of pesticides (atrazine, azoxystrobin, bentazon, -cyhalothrin, penoxsulam, and terbuthylazine) from aquatic worms (*Nereis diversicolor*) and bivalves (*S. plana*), Rodrigues and coworkers [63] preferred the use of lower extraction temperatures and Florisil as a fat retainer. In particular, a temperature of 40 °C was set during the extraction performed with methanol and exploiting the capabilities of Florisil, loaded directly into the extraction cell along with the sample, to retain fats and interferents. S-PLE was followed by on-line SPE–ultra-performance liquid chromatography (UPLC)-ESI-MS/MS.

PLE has also been used as an alternative to conventional sample preparation QuEChERS for the analysis of multiresidue pesticides in tuber matrices, followed by determination by GC-MS/MS. In this case, ethyl acetate and 100 °C were used during the extraction, obtaining recoveries in the 70–120% range for all the 150 pesticides analyzed [45].

The extraction of pesticides from plant material can be complicated by the presence of pigments, which must be removed during sample preparation together with the fat. For this purpose, Raina-Fulton et al. (2018) [46] and Kinross et al. (2020) [47] proposed two different S-PLE approaches. The first one used C18, Florisil, and a synthetic carbon (Anasorb) directly loaded into the extraction cell along with the sample (Matcha) to be subjected to extraction with ethyl acetate at 100 °C [46]. The second one used layers of Florisil and graphitized carbon black (GCB) in the extraction cell along with the sample (alpha alpha and citrus leaves), followed by extraction with a mixture of ethyl acetate/hexane (25/75, *v*/*v*) at 80 °C. Comparable results in terms of performance of the method were obtained by using the Energized Dispersive Guided Extraction (EDGE^®^) instrument, introduced by CEM Corporation in October 2017 and developed to combine PLE with dispersive-SPE (dSPE) [47].

#### 5.2.2. Veterinary Drugs and Metabolites

PLE has been widely used for the extraction of veterinary drug residues and their metabolites from different food matrices of animal origin. In addition to sample extraction, carried out not only with different solvent mixtures but also with pure water, a purification step of the extract, either in-cell (S-PLE) or separate (SPE, GPC), is necessary before analytical determination. The food analyzed undergoes pretreatment such as grinding, drying, freeze-drying, and subsequent homogenization before being loaded into the extraction cells, mostly after mixing with dispersing agents (DE, sand) and any effective adsorbent phases to retain fats and interferents.

Specific PLE-based methods were optimized for the extraction of tetracyclines (TCs), antibiotics from various animal products (chicken eggs and meat, fish, shrimp, clams, and pork), with the aim of reducing the laboriousness of conventional sample pretreatments. During the optimization of these methods, special attention was paid to instrumental parameters (time, temperature, pressure) because of the unstable nature of tetracyclines, which decompose rapidly under the influence of light and atmospheric oxygen, forming more than 14 different degradation products. In detail, the method proposed by Liu and coworkers [49] applied an extraction with acidified (pH 4) methanol/trichloroacetic acid (TCA) (3/1, *v*/*v*) at 60 °C and a pressure of 65 bar. The extract thus obtained appears to be so clean that it does not require further purification before the HPLC-UV determination. In contrast, the method proposed by Jiao and collaborators [54] exploits the potential of the S-PLE. In particular, they first applied copper(II) isonicotinate as an on-line clean-up sorbent, which is characterized by a large surface area, specific pore size, and hydrophilic properties giving it the ability to retain interfering compounds (substantially fatty acids) compared to other clean-up sorbents (Florisil, silica gel, and alumina). The highest extraction efficiency was obtained at 70 °C and using a static extraction time of 15 min with methanol. Furthermore, in 2020, Wang and coworkers [62] developed a multi-residue method for simultaneous analysis of tetracyclines and *β*-receptor agonists in chicken by coupling PLE with ultra-high-performance liquid chromatography–tandem mass spectrometry (UHPLC-MS/MS). The extraction temperature was set at 60 °C, as it represented the best compromise between recoveries and co-extraction of interferents (soluble organic matter, such as proteins). During the optimizations, the authors focused on selecting the most appropriate solvent mixture. Therefore, initially, the sample was cleaned-up with hexane to remove lipids, and the extract was discarded; then, the analytes of interest were extracted with methanol/acetonitrile (1/1, *v*/*v*), and the collected extracts were treated with further acetonitrile to remove residue proteins. In this way, better extraction performances were obtained in comparison with those achieved by using QuEChERS and ultrasound-assisted extraction. The same working group, in the previous year, used PLE for the simultaneous extraction of six steroid hormones and six antibiotics in fatty foods by using a mixture of the three selected solvents, hexane/acetonitrile/methanol (2/1/1, *v*/*v*/*v*) at 60 °C, without further clean-up prior to the UHPLC-MS/MS analysis [61].

Only two of the methods reported in Table 3 for the analysis of veterinary drug use water as the extraction solvent during PLE. Both resort to the purification of the extract on SPE before analytical determination by LC-MS. The first applied 100 °C for 10 min to freeze-dried mussels previously mixed and homogenized with Ottawa sand [56]. The second method applied 100 °C for 15 min to fish samples previously ground and blended with sea sand [59].

PHWE has been used for the analysis of trace amphenicols in poultry tissues. This approach is based on the use of water as an extraction solvent at temperatures between 100 and 374 °C (critical point of water at 22 MPa) and at a sufficiently high pressure to keep it in a liquid state. Under these subcritical conditions, the chemical properties of water (polarity, viscosity, surface tension, and dielectric constant) are modified, becoming more like those of an organic solvent. However, a risk inherent to the use of subcritical water as an extraction solvent is that it could decompose those compounds that are thermolabile and/or prone to hydrolytic attack. Therefore, the optimization of the temperature is a crucial point. Basification of pure water with ammonium hydroxide (0.2%) increased the recovery of some weak base metabolites (florfenicol amine) [55].

In addition to the method already described for tetracycline analysis using copper(II) isonicotinate [54], the other methods resorting to the S-PLE have used neutral aluminum oxide [48,50,51,52,60] directly loaded in the extraction cells along with the sample. In all cases, despite this in-cell clean-up, further purification steps were performed, such as SPE or GPC, prior to the LC-MS determination.

A purification system without the use of adsorbent materials placed in the extraction cell consists of performing an initial washing of the sample with an apolar solvent, such as hexane, capable of eliminating lipids and more, before the real extraction of the analytes of interest with polar or medium-polar solvents [53,62].

**Table 4 foods-12-02017-t004:** Application of PLE for the analysis of different contaminants in food.

Analytes	Matrix	Sample Pre-Treatment	Cell Preparation	Extraction Solvent	In Cell Clean-Up	T (°C)/P	Time/No. Cycles/Flush Volume	Purge with Nitrogen	Sample Post-Treatment	Analysis	Ref.
Aflatoxins and ochratoxin A	Dried fruits	Finely blended and homogenized with water at a ratio of 2:1	4 g of sample mixed with 1 g of DE in a 3 mL cell and	Water/methanol (70/30, *v*/*v*)	no	110/1500 psi	5 min/3 cycles/50%	50 s	On-line SPE (Strata C18-E)	UHPLC-MS/MS	[64]
Alternaria mycotoxins	Tomato	Blended	5 g of sample in a 22 mL cell mixed with 5 g of DE	Methanol/water (25:75, *v*/*v*)	no	70/1000 psi	5 min/1 cycle/60%		MISPE	HPLC-FLD and UPLC-MS/MS	[65]
Mycotoxins	Staple cereals	Milled with a mechanical blender	4 g of sample in a 20 mL cell mixed with 3 g of DE	Water/ethanol (55/45, *v*/*v*)	no	162/1000 psi	10 min at 5 mL/min			HPLC-MS/MS	[66]
Paraben-type preservatives and benzophenone-type ultraviolet light filters	Plant and animal seafood	Homogenized	1 g of sample in a 22 mL cell mixed with DE	Methanol	no	70/1500 psi	3 min/1 cycle/60%	60 s	Mixed mode cationic exchange SPE	LC–QqLIT-MS/MS	[67]
Ethyl carbamate	Fermented solid foods	Ground	5 g of sample in a 34 mL cell with 5 g of Florisil	Ethyl acetate	yes	50/10.3 MPa	5 min/2 cycles/60%	120 s	Evaporation to dryness and dissolution in 1.0 mL of methanol	GC-MS/MS	[68]
MCPDEs and Ges	Bread and rolls, fine bakery wares, smoked fish products, fried and roasted meat, potato-based snacks and fried potato products, cereal-based snacks, and margarines	Immersed in liquid nitrogen and ground	5 g of sample in a 33 mL cell mixed with 5 g of polyacrylate and 15 g of sand	tert-butyl methyl ether	no	40/103 bar	5 min/2 cycles/100%	60 s	Derivatization with phenyl boronic acid	GC-MS	[69]

### 5.3. Other Food Contaminants

The potential of the PLE technique has also been exploited for the analysis of mycotoxins in foods of plant origin. Rico-Yuste and coworkers [65] developed a method for the extraction of *Alternaria* mycotoxins from tomato samples using methanol/water (25/75, *v*/*v*) at 70 °C. Particular attention was paid to the optimization of the temperature, which strongly influenced the solubility of the mycotoxins. The extracts were purified through molecularly imprinted solid phase extraction (MISPE) cartridges prior to the LC with fluorescence detector (FLD) determination. Later, Gbashi and collaborators [66] succeeded in analyzing 15 different mycotoxins in staple cereals by using a laboratory-scale PHWE, with water/ethanol (55/45, *v*/*v*) at 162 °C, prior to the LC-MS/MS determination. This approach has been considered a valid green alternative to the conventional solvent-based extraction methods.

A simultaneous determination of aflatoxins and ochratoxin A in dried fruits has been proposed by Campone et al. [64], using PLE followed by on-line SPE-UHPLC-MS/MS. The PLE was performed with water/methanol (70/30, *v*/*v*) at 110 °C.

Parabens (PBs) and benzophenone-type ultraviolet light filters, widely used in personal care products and considered potential EDCs, have been analyzed in seafoods, including those of plant and animal origin. PLE was chosen for the sample extraction, and an additional SPE clean-up was performed to minimize matrix effects and improve the sensitivity. Analyte identification and confirmation were performed using liquid chromatography quadrupole linear ion trap-tandem mass spectrometry (LC-QqLITMS/MS). In this case, PLE turned out to be a relatively green and efficient approach with low solvent and time consumption compared with other conventional methods such as Soxhlet or ultrasound-assisted extraction [67].

Ethyl carbamate, which is present in many fermented foods and alcoholic beverages and is potentially toxic and probably carcinogenic to humans, was analyzed by Liao and coworkers [68] using S-PLE followed by GC-MS/MS. Usually, its low content, matrix complexity, and contamination of the inlet and column by high-boiling matrix compounds during GC analysis are the main obstacles to the determination of ethyl carbamate in fermented foods. Therefore, time-consuming sample extraction and clean-up is required, resulting in multiple operation steps, which may compromise recoveries. The use of S-PLE technique was proposed to reduce the exhaustive extraction and post-cleaning procedures. In particular, Florisil was used as the clean-up sorbent and extraction was performed with ethyl acetate at 50 °C. Thus, recoveries ranged from 98 to 107% with a relative standard deviation of less than 7%.

Growing concerns about possible health risks from the presence of fatty acid esters of 3-monochloro-1,2-propanediol and 2-monochloro-1,3-propanediol (MCPDE) and 2,3-epoxide-1-propanol or glycidol (GE) in processed food oils and fats have led to the development of a number of analytical methods for their determination. Among them, the extraction of fats and analytes from food samples by PLE was studied and optimized on Belgian waffle samples by Samaras and coworkers [69]. The extraction was performed with tert-butyl methyl ether at 40 °C, followed by derivatization and GC-MS determination. The optimized method was then successfully applied for the analysis of target compounds in more than 650 different food samples belonging to different categories: bread and rolls, fine bakery wares, smoked fish products, fried and roasted meat, potato-based snacks, and fried potato products, cereal-based snacks, and margarines.

## 6. Conclusions

This review was intended to provide a general overview of the principles, main factors, and some applications of PLE for the analysis of various food contaminants. PLE is an advanced sample preparation technique that is very useful for the determination of environmental contaminants in solid or semi-solid matrices. It has the advantage of being automated, permitting serial extractions, saving time and solvent consumption. Moreover, PLE generally uses less solvent volume than conventional methods such as liquid–liquid, solid–liquid, and Soxhlet extraction. Complementary steps, such as SPE or GPC, are commonly employed to concentrate/purify the extract. However, the continuous evolution of this technique has also led to the development of S-PLE, which allows purification directly in the cell using one or more adsorbent materials, avoiding further sample purification.

A quick look at the recent development of coupling PLE with separation techniques to reduce sample preparation steps and to meet high throughput requirements in control laboratories is also provided.

In addition, applications from the past 10 years covering a wide range of contaminants in food samples were selected, summarizing the parameters and conditions used.

The wide range of application, its flexibility, and good performance in terms of extraction efficiency have led to widespread use of this technique, not only at the level of routine laboratory analysis but also with great potential at the industrial level.

## Figures and Tables

**Figure 1 foods-12-02017-f001:**
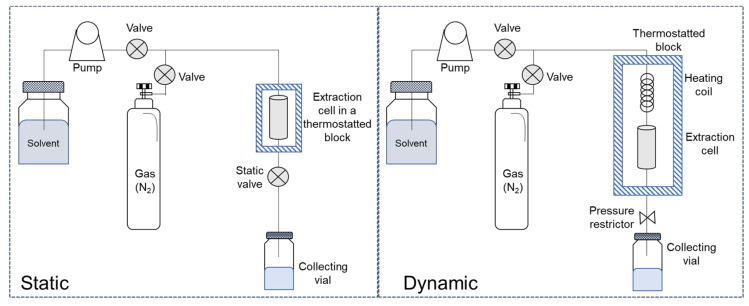
Schematic PLE system configuration for static and dynamic procedures.

**Table 1 foods-12-02017-t001:** Comparison of the most commonly used extraction methods for solid samples.

	Solvent Consumption	Process Time	Sample Post-Treatment	Advantages	Disadvantages
PLE	Low	Short	Yes/no (in cell clean-up)	Possibility to perform more extraction cycles; high sample throughput	High instrumentation cost; long cell preparation
MAE	Low	Short	Yes	Very high sample throughput, rapid heating, combination with hydrolysis and derivatization chemistry	Limited choice of the extraction solvent; filtration is needed; high instrumentation cost
SoxE	Medium	Very long	Yes	Solvent recycling; require cheap apparatus	Low sample throughput
USAE	Medium	Short	yes	Requires cheap apparatus	Filtration is needed
SFE	None/very low	Medium	Usually not required	Direct coupling with analytical determination	Not useful for extracting ionic and polar compounds; high instrumentation cost
SLE	High	Long	Yes	Does not require special apparatus	Low sample throughput

SoxE: Soxhlet; USAE: ultrasound-assisted extraction; SFE: supercritical fluid extraction; SLE: solid–liquid extraction.

## Data Availability

No new data were created or analyzed in this study. Data sharing is not applicable to this article.

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
