# Peer review of "Pressurized Liquid Extraction: A Powerful Tool to Implement Extraction and Purification of Food Contaminants"

_foods, 2023, doi:10.3390/foods12102017_

Round 1
Reviewer 1 Report
The review carried out is relevant, novel and of interest to professionals in the study of analytes from various samples. In general, the information presented is adequately organized and the information in the tables is clear.
Some suggestions to improve the manuscript are the following:
1. Throughout the manuscript the advantages of PLE extraction are presented with reference to traditional and modern extraction methods, such as liquid-liquid, soxhlet extraction, ultrasonic extraction, microwave assisted extraction, SPE purification among others. I believe that this information can be represented in a comparative table or in a graphic organizer.
2. Page 4, line 143 please specify the meaning "DE"
3. In references section, verify that you use the abbreviated journal name, as specified in the instructions for authors.
Author Response
The review carried out is relevant, novel and of interest to professionals in the study of analytes from various samples. In general, the information presented is adequately organized and the information in the tables is clear.
The authors appreciate the consideration given by the Reviewer #1 to the present manuscript.
Some suggestions to improve the manuscript are the following:
- Throughout the manuscript the advantages of PLE extraction are presented with reference to traditional and modern extraction methods, such as liquid-liquid, soxhlet extraction, ultrasonic extraction, microwave assisted extraction, SPE purification among others. I believe that this information can be represented in a comparative table or in a graphic organizer.
A brief comparison between PLE and both traditional and modern extraction methods for solid samples has been reported in Table 1.
- Page 4, line 143 please specify the meaning "DE"
The text has been modified.
- In references section, verify that you use the abbreviated journal name, as specified in the instructions for authors.
The Reference section has been modified according to the instructions for authors.
Reviewer 2 Report
The revised manuscript is interesting; however, it is required to review the following:
Line 16,35: semisolid or semi-solid like in line 12 (it is necessary to homogenize terms throughout the document)
Line 53: consumption [insert reference].
Line 58: insert space… 200 °C
Line 63-74: references missed ?
Line 107: 3.1. On
Line 119: insert space…150 °C
Line 123: insert space… 200 °C
Line 130: remove italic text format from 4.
Line 165-169: references missed ?
Line 181: insert space… temperature [13].
Line 186: 4.4. Extraction
Line 187-198: Could you support the information with the use of an additional reference?
Line 199: 4.5. Effect
Line 204: ….temperature [insert reference].
Line 209: 4.6. Selective
Line 236: 5.1.1. EDCs
Line 293: correct 34 mL
Line 293: correct 33 mL
Line 293: correct 100 mL
Line 293: correct 22 mL
Line 293: correct 11 mL
Line 293: correct 10 mL
Line 293: correct 66 mL
Line 293: correct 0.01-0.03
Line 293: scientific names should be written in italic text (Mirounga leonine; Arctocephalus gazella)
Line 300: insert space 70 °C
Line 304: 5.1.2. Hydrocarbon
Line 314,316,333,337,398,411,418: insert space 100 °C
Line 346: insert space 75 °C
Line 355: 230 °C
Line 356: 200 °C
Line 357: 80 °C
Line 360: 5.1.3. FRs
Line 367: 150 °C
Line 378,420: 80 °C
Line 385,490: remove the middle hyphen from (number mL)
Line 405: 40 °C
Line 415: use the correct format for citing references
Line 424: 5.2.2. Veterinary
Line 439: 14 different ? like in line 501 (check from which number is written with letter or numbering)
Line 441,452,462: 60 °C
Line 448,497: 70 °C
Line 450: β-receptor
Line 466,468,471: 100 °C
Line 472: 374 °C
Line 502: 162 °C
Line 507: 110 °C
Line 526: 50 °C
Line 534: 40 °C
Reference section
- Titles must be written in lowercase text format, except for the first letter of the first word.
- A dot must be inserted in each abbreviated word of the journal of the reference
- review the Microsoft word template to verify the correct format of this section
- Correct format text of scientific names
Author Response
The revised manuscript is interesting; however, it is required to review the following:
The authors appreciate the consideration given by the Reviewer #2 to the present manuscript.
Line 16,35: semisolid or semi-solid like in line 12 (it is necessary to homogenize terms throughout the document)
The term has been homogenized throughout the manuscript.
Line 53: consumption [insert reference].
A reference has been added.
Line 58: insert space… 200 °C
The space before “°C” has been added throughout the manuscript.
Line 63-74: references missed ?
Reference have been added.
Line 107: 3.1. On
The text has been modified.
Line 119: insert space…150 °C
Line 123: insert space… 200 °C
The space before “°C” has been added throughout the manuscript.
Line 130: remove italic text format from 4.
The text has been modified.
Line 165-169: references missed ?
More references have been added to the section 4.2.
Line 181: insert space… temperature [13].
The text has been modified.
Line 186: 4.4. Extraction
The text has been modified.
Line 187-198: Could you support the information with the use of an additional reference?
More references have been added to the section 4.4.
Line 199: 4.5. Effect
The text has been modified.
Line 204: ….temperature [insert reference]
A reference has been added.
Line 209: 4.6. Selective
The text has been modified.
Line 236: 5.1.1. EDCs
The text has been modified.
Line 293: correct 34 mL
Line 293: correct 33 mL
Line 293: correct 100 mL
Line 293: correct 22 mL
Line 293: correct 11 mL
Line 293: correct 10 mL
Line 293: correct 66 mL
The text in all tables has been modified accordingly.
Line 293: correct 0.01-0.03
The text has been modified.
Line 293: scientific names should be written in italic text (Mirounga leonine; Arctocephalus gazella)
The text has been modified.
Line 300: insert space 70 °C
The space before “°C” has been added throughout the manuscript.
Line 304: 5.1.2. Hydrocarbon
The text has been modified.
Line 293: correct 0.01-0.03
The text has been modified.
Line 314,316,333,337,398,411,418: insert space 100 °C
Line 346: insert space 75 °C
Line 355: 230 °C
Line 356: 200 °C
Line 357: 80 °C
The space before “°C” has been added throughout the manuscript.
Line 360: 5.1.3. FRs
The text has been modified.
Line 367: 150 °C
Line 378,420: 80 °C
The space before “°C” has been added throughout the manuscript.
Line 385,490: remove the middle hyphen from (number mL)
The text has been modified.
Line 405: 40 °C
The space before “°C” has been added throughout the manuscript.
Line 415: use the correct format for citing references
References in the correct form have been added.
Line 424: 5.2.2. Veterinary
The text has been modified.
Line 439: 14 different ? like in line 501 (check from which number is written with letter or numbering)
The text has been modified.
Line 441,452,462: 60 °C
Line 448,497: 70 °C
The space before “°C” has been added throughout the manuscript.
Line 450: β-receptor
The text has been modified.
Line 466,468,471: 100 °C
Line 472: 374 °C
Line 502: 162 °C
Line 507: 110 °C
Line 526: 50 °C
Line 534: 40 °C
The space before “°C” has been added throughout the manuscript.
Reference section
- Titles must be written in lowercase text format, except for the first letter of the first word.
- A dot must be inserted in each abbreviated word of the journal of the reference
- review the Microsoft word template to verify the correct format of this section
- Correct format text of scientific names
The Reference section has been totally revised and modified accordingly.
Reviewer 3 Report
I have recorded my comments on various pages in the attached PDF, which are self-explanatory.
Few general suggestions are below:
Abstract: By the look of whole review paper, it is suggested to please include few more sentences in the Abstract, which can be easily extracted.
Introduction: Few paragraphs comprised on only one/two sentences. Please amalgamate/move/merge them with the most relevant paragraphs.
Conclusion: The Conclusion section is too short. The author may include few sentences regarding PLE's introducing concepts. The conclusion may also include few sentences related to the main ideas of major topics included in the review paper.

Author Response
I have recorded my comments on various pages in the attached PDF, which are self-explanatory.
The text has been modified accordingly.
Few general suggestions are below:
Abstract: By the look of whole review paper, it is suggested to please include few more sentences in the Abstract, which can be easily extracted.
The abstract has been lengthened, but only one sentence could be added due to the 200-word maximum limit imposed by the journal guidelines.
Introduction: Few paragraphs comprised on only one/two sentences. Please amalgamate/move/merge them with the most relevant paragraphs.
The introduction has been revised with the aim of merging stand-alone sentences with the most relevant paragraphs.
Conclusion: The Conclusion section is too short. The author may include few sentences regarding PLE's introducing concepts. The conclusion may also include few sentences related to the main ideas of major topics included in the review paper.
The Conclusion section has been improved trying to recall all the points examined during the review.